# The Role of Phonological Awareness, Pinyin Letter Knowledge, and Visual Perception Skills in Kindergarteners’ Chinese Character Reading

**DOI:** 10.3390/bs12080254

**Published:** 2022-07-27

**Authors:** Han Yuan, Eliane Segers, Ludo Verhoeven

**Affiliations:** 1Faculty of Education, Shenzhen University, Shenzhen 518061, China; 2Behavioural Science Institute, Radboud University, Montessorilaan 3, 6525 HR Nijmegen, The Netherlands; eliane.segers@ru.nl (E.S.); l.verhoeven@pwo.ru.nl (L.V.)

**Keywords:** phonological awareness, visual perception skills, Pinyin letter knowledge, Chinese character reading

## Abstract

Word identification models assume that words are identified by at least two sources of information and analysis; one is phonological, and the other is visual. The present study investigated the influence of phonological awareness, Pinyin letter knowledge, and visual perception skills on Chinese character recognition after controlling for vocabulary, rapid naming, and verbal short-term memory in 80 Mandarin-speaking kindergarten children. Children were tested on phonological awareness (syllable awareness, onset-end rhyme awareness, and tone awareness), Pinyin letter naming, and visual perception (visual discrimination and visual-spatial relationships). The results showed that variance in Chinese character recognition could be explained by syllable awareness and tone awareness, but not by visual perception skills or Pinyin letter knowledge. Analyses further indicated that Pinyin letter knowledge moderated the relationship between tone awareness and Chinese character recognition. A focus on tone awareness and syllable awareness in the kindergarten may help Chinese children to accomplish the transition from phonological awareness to early literacy, while Pinyin letter knowledge can help children to make the connection between Chinese speech and writing.

## 1. Introduction

Different models of word identification during reading assume that words can be identified by at least two sources of information and analyses: phonological and visual [1,2,3,4]. Phonological awareness, letter knowledge, rapid naming, and short term memory have been shown to be important phonological predictors of learning to read across alphabetic orthographies [5]. Studies have also reported that visual skills are generally correlated with reading ability [6,7,8,9] and children rely on partial visual information for word recognition in their initial step of learning to read [10].

While the focus of reading research has mostly been on alphabetic scripts [11], studies on learning to read in Chinese have also emerged [12,13]. Empirical studies and neuroimaging evidence have shown that for early Chinese character reading, phonological awareness, rapid naming, and short term memory are important predictors, as in alphabetic scripts [14,15,16]. Other factors, more specific for Chinese character reading, have also been identified, such as Pinyin, which is taught in mainland China but not Hong Kong or Taiwan [17]. However, no study, so far, has included phonological awareness, Pinyin letter knowledge, and visual perception skills in one design, leaving open the question of which is more relevant for Chinese character reading for young children, especially for children who have not received formal literacy education. In the present study, we examined this question in a group of 80 Mandarin-speaking kindergarten-3 children in mainland China.

### 1.1. Phonological Awareness as Predictor for Character Reading

Phonological awareness, which refers to the ability to perceive and manipulate the small units of sounds in words [18], is key to learning to read across orthographies [19,20,21]. Phonological awareness is not only important in learning to read in English, which has an opaque orthography, but also in other alphabetic languages, with more transparent orthographies [18,22,23,24]. In alphabetic languages, phonological awareness develops from larger to smaller sound units, that is, from syllables to rhymes to phonemes [21,25]. Studies have revealed that the awareness of phoneme-size units was the most powerful predictor in alphabetic language reading [22].

In previous research, four aspects of Chinese phonological awareness have been identified: syllable awareness, onset awareness, rhyme awareness, and tone awareness. These four aspects have been proven to be related to learning to read Chinese. Regarding syllable awareness, unlike alphabetic languages, Chinese is divisible at the syllable level [26,27], which means the basic speech unit of Chinese is the syllable and each syllable can be divided into two parts: the onset and rhyme [28]. In addition, Chinese is a tonal language. A syllable cannot be comprehended unless a certain tone is added on it. Mandarin Chinese tone patterns vary in one of four ways, (1) high level, (2) rising, (3) low falling and rising, and (4) high falling. Chinese children become aware of large sound units before smaller ones. Syllable awareness develops first, which is followed by rhyme and tone awareness. Onset awareness develops last [14,29].

Both concurrent and longitudinal studies have shown that syllable awareness significantly predicts Chinese word reading [30,31]. For example, McBride-Chang and Kail [30] compared the relationship between four cognitive skills with word recognition in Hong Kong kindergarten students and showed that syllable deletion was the strongest predictor of Chinese reading. McBride-Chang and Ho [32] again found that syllable deletion accounted for unique variance in Chinese word recognition in a 2-year longitudinal study of Hong Kong kindergarteners who learned English as a second language.

Onset awareness is the second phonological predictor that may be important for Chinese character reading. Children usually receive Pinyin instruction at the beginning of grade 1 in primary school and they are taught how to dissect syllables into initial and final sound segments (the onset and the rime). For example, the character “三” (three) is pronounced as “s-an”, “s” is the onset and “an” is the rime. The study of McBride-Chang et al. [27] showed that onset awareness in Chinese is positively associated with Chinese character recognition in mainland China kindergarten and primary school participants.

Rhyme awareness may also be a correlate to Chinese reading. Huang and Hanley [33] examined the predictive power of phonological awareness (rhyme-alliteration recognition task) among first graders in Taiwan and found that the level of phonological awareness before learning the alphabetic system significantly related to character reading at the end of the first year. Ho [34] found that the phonological scores (rhyme detection and memory for non-rhyming Chinese characters) correlated significantly with the character reading scores among 47 Chinese second graders. Li et al. [35] also proved that rhyme awareness was a unique predictor of character reading in primary school children.

Finally, tone awareness is considered a relevant phonological skill for character recognition. The existence of many homophonic written characters in Chinese script, such as “北” /bei/ 3 and “贝” /bei/ 4 (the numbers mean different tones), may cause the discrimination of tonal information to help children to recognize characters. Moreover, some homophonic written characters are included in a phonetic radical neighborhood and are visually similar, such as the following three characters “妈”, “马”, and “骂”. All of them are pronounced as /ma/. However, the former character means “mother” and is pronounced /ma/ with the first tone, the middle one means “horse” and is pronounced /ma/ with the third tone, and the latter one means “curse” and is pronounced /ma/ with the fourth tone. Another reason why tone awareness is so important could rely on the fact that Mandarin Chinese has some polyphones. For example, “背” is pronounced as /bei/ 4 which means back, it is pronounced as /bei/ 1 as well and means bear. Tone discrimination and production were found to correlate significantly with Chinese word reading [36]. Shu et al. [14] also demonstrated that tone awareness emerged as the strongest correlate of reading for Mandarin-speaking kindergarten children.

### 1.2. Pinyin Knowledge, Phonological Awareness, and Chinese Reading

In alphabetic languages, phonological awareness is bi-directionally linked with letter knowledge [37,38]. Letter knowledge assists children to establish and recall words from memory, and to decode unfamiliar words [39].

Chinese is a nonalphabetic language as it is morphosyllabic in its representation of characters. Pinyin, which is an alphabetic system, is introduced in mainland China to help children to learn the spelling of Chinese characters. Pinyin is the Chinese phonetic alphabet and it is a part of the national curriculum for learning to read in mainland China. Because Chinese characters cannot be pronounced by recourse to grapheme-phoneme correspondence rules, Pinyin is used as a tool for children to connect oral Chinese and printed symbols [40]. Pinyin enables children to derive the meanings of characters that are visually unfamiliar but auditorily familiar [41]. In other words, Pinyin knowledge may provide a bridge between speech and writing in Chinese. Those children with Pinyin knowledge may be able to make the connection between speech and writing, while this is more difficult for those without Pinyin knowledge.

Studies have used various tasks to test Pinyin knowledge and showed a positive association between Pinyin knowledge and character reading. For example, Wang et al. [42] found that after age, kindergarten level, nonverbal reasoning statistically controlled, Pinyin letter name knowledge, morphological awareness, and rapid naming uniquely predicted Chinese word reading in a combined K-2 and K-3 group. For K-3 Chinese monolingual children, invented Pinyin spelling was found to be a strong predictor of Chinese word reading one year later, even with syllable deletion, phoneme deletion, and letter knowledge that were tested in Time 1 controlled [17]. Li et al. [43] followed grade 1 children in Beijing for one year. They showed that Pinyin sentence reading had a direct impact on children’s later literacy skills. A cross-language study found that invented Pinyin spelling was highly predictive of subsequent Chinese word reading for grade 2 English-Chinese bilingual children [40].

In the previous studies, Pinyin skill was found to be positively related to phonological awareness [14,44,45]. However, the role of Pinyin knowledge, especially Pinyin letter knowledge, in the relation between phonological awareness and character reading for kindergarten children is unclear. Only the study of Li et al. [43] showed that an advanced Pinyin knowledge and Pinyin sentence reading, mediated the relationship between phonological awareness and later literacy skills which involve sentence reading, vocabulary, and character writing for primary school children.

### 1.3. Visual Skills As Predictor for Character Reading

Visual skills have been proved to be important in learning to read in alphabetic languages. One study showed that prereading performance in spatial relationships, position in space, and figure-ground perception were significant predictors of later English word reading even after controlling for IQ and age in English children [46]. Franceschini et al. [47] also suggested that core visual processing skills such as visual spatial attention in young prereaders is a causal factor in subsequent reading acquisition after controlling for age, nonverbal IQ, speech-sound processing, and nonalphabetic cross modal mapping.

Two main dimensions of contrast between alphabetic languages and Chinese have been pointed out, one at the script level, and one at the mapping principle level. At the script level, Chinese has a rectangular layout of its graphic components. Components of each character are arranged side by side, top to bottom, or inside–outside [48]. For example, in the Chinese character 妈 ‘mother,’ the components are arranged side by side. This character 妈 is composed of the radical 女 ‘female’ and another symbol 马 that represents the segmental string or syllable /ma/. Furthermore, each Chinese character usually consists of radicals and radicals consist of different numbers of strokes. Before receiving formal reading education, normally mainland China kindergarten children use the “Look and Say” strategy to learn Chinese characters more or less during their daily life or at home. The complexity of Chinese orthography [49] and the learning strategy has led researchers to hypothesize that visual skills are critical for Chinese reading of young children. Various tasks were used to test these visual skills, such as visual perception, visual memory, visual-verbal association, and visual-orthographic processing skills.

Studies focusing on the relation between visual skills and Chinese character reading had mixed results. Several studies found that visual skills were positively related to character recognition. McBride-Chang et al. [50] tested kindergarten children twice in a longitudinal study in Hong Kong and mainland China. They found visual spatial relationships predicted Chinese character recognition at Time 1 among Hong Kong children and in mainland China children nine months later. Luo et al. [51] tested the role of two visual processing skills in learning to read Chinese characters in kindergarten children and primary school children in Beijing: geometric-figure processing and character-configuration processing. Geometric-figure processing predicted reading only in kindergarten while character-configuration processing explained unique variance in reading in both kindergarten and primary grades. Anderson et al. [52] found that visual-orthographic processes play a significant role in the literacy development of Chinese primary children. Siok and Fletcher [41] found that visual form constancy and visual sequential memory predicted character reading in grades 1 and 2 in Beijing Chinese children after controlling for age and IQ. Kuo et al. [53] showed that visual analogical skill made a significant contribution to the acquisition of characters varying in properties, regardless of age. For Taiwan and Hong Kong third-graders, visual paired associate learning was significantly related to reading ability [33].

Some studies did not find evidence for the role of visual skills in character recognition. For example, a study conducted by McBride-Chang and Kail did not find a direct association between visual processing and character reading in a combined Hong Kong kindergarten group [30]. Li et al. [35] examined metalinguistic and cognitive skills in relation to Chinese character recognition for mainland children in kindergarten and primary school. Only syllable deletion, morphological construction, and rapid number naming were found to have unique correlates of Chinese character recognition in kindergarteners. Visual-orthographic skills were not uniquely associated with Chinese character reading at any grade level.

### 1.4. Present Study

Phonological awareness, Pinyin knowledge, and visual skills have been related to Chinese character reading separately. However, no study has combined phonological awareness, Pinyin letter knowledge, and visual perception skills to consider their unique contribution to explaining variation in Chinese character recognition for kindergarten children.

The aim of the present study was to investigate first whether visual perception skills would predict Chinese character recognition over and above phonological awareness and Pinyin letter knowledge, and the other way around, as no clear hypothesis on this could be derived from the literature. To test the unique variance in Chinese character recognition predicted by these three skills, we included two phonological processing tasks: rapid naming and short term memory to control for their contribution. We expected both phonological awareness and visual perception skills to predict Chinese character recognition, and also to find an effect of Pinyin letter knowledge. Second, we examined the role of Pinyin letter knowledge in the relationship between phonological awareness and character reading, to find out whether Pinyin knowledge can be seen as a bridge between speech and writing in Chinese.

## 2. Method

### 2.1. Participants

A total of 80 (39 boys; 41 girls) Mandarin-speaking K3 children (third-year kindergarteners) in Mainland China participated in the present study. The mean age was 67 months. In Mainland China, kindergarten is a three-year independent educational stage. Children receive formal literacy instruction from the first grade, so participants had not received formal literacy instruction when they were tested. However, children were exposed to Chinese characters in kindergarten through activities such as story books reading, learning corner activities, and group activities. Story books for kindergarten children in mainland China consist of a lot of pictures and few accompanying words to describe the stories. In story book reading activities, children sit together in the middle of the room, teachers always put the title and pictures of the stories on the screen, children look at the screen and speak out the title by following their teachers. However, teachers do not teach children how to spell the Pinyin of the words or how the words are constructed. The purpose is to let children know the brief idea of the stories. Children may also get knowledge of Pinyin and characters by the influence of daily life, such as public media and parental education. All the participants were judged to be typically developing according to the teachers’ reports.

### 2.2. Procedure

All the participants were randomly chosen from three different classrooms of a kindergarten affiliated with a university in Central China. A consent form from the parents and the kindergarten was obtained before the tests and a convenient time was arranged for each participant according to the kindergarten activity schedule. Children were tested individually in a quiet room in the kindergarten. All the tasks were conducted by the first author and four trained preschool education master students who were native Mandarin Chinese speakers. Finishing all the tasks took about 1 h for each child.

### 2.3. Materials

#### 2.3.1. Chinese Receptive Vocabulary

Chinese receptive vocabulary was assessed using the Chinese Peabody Picture Vocabulary Test [54]. The Chinese version is based on the Peabody Picture Vocabulary Test—revised from the English version [55]. There are five practice items and 125 test items. In each item, children were presented with four black and white pictures and asked to point out the correct picture corresponding to the target words which were named by the experimenter. The beginning test set was decided by the age of children, the stop set was the set in which children made six mistakes in eight consecutive items, and the fundamental set was the highest set in which children gave the correct answers in eight consecutive items.The score was the number of the right answers from the foundation set to the stop set. The Cronbach’s alpha of this task in the current sample was 0.95, which is considered excellent internal consistency.

#### 2.3.2. Rapid Naming

Rapid naming was measured by a rapid word naming test, which was developed for the current research project based on a Dutch version of rapid word naming [56]. Five one-syllable Chinese words with their corresponding pictures were selected: 脚 ([jiǎo], feet), 鱼 ([yú], fish), 树 ([shù], tree), 伞 ([sǎn], umbrella), 包 ([bāo], bag) and arranged in five columns of ten pictures each according to the order of the Dutch version. These words were both easy and highly familiar to young children. Before the beginning of the task, children had to identify all the five pictures orally one by one and the experimenter made sure each child knew the correct words that corresponded to the pictures. Then the children were asked to name the pictures in the columns as quickly and correctly as possible in one minute. The number of the correctly named pictures that the children gave was the final score. The maximum total score is 50. The Cronbach’s alpha of this task in the current sample was 0.95.

#### 2.3.3. Short Term Memory

Short term memory was measured by a Chinese word span task which was also developed for the current research project and based on a Dutch word span task [57]. Seven one-syllable words in Chinese were chosen: 车 ([chē], car), 马 ([mǎ], horse), 人 ([rén], human), 手 ([shǒu], hand), 花 ([huā], flower), 一 ([yī], one), and 刀 ([dāo], knife). They were both easy and highly familiar to the children. Twelve sequences were constructed which consisted of these words according to the order of Dutch task. The sequences varied in length from two to seven words. The experimenter spoke out loud each sequence of words in a calm tempo and without prosodic cues and children were asked to repeat. The Cronbach’s alpha of this task in the current sample was 0.56. It is not surpring that the task has a low reliability, as 68 children obtained high scores in this task (score ≥ 10, total score = 12), so the task had a restricted spread of scores.

#### 2.3.4. Chinese Phonological Awareness

Chinese phonological awareness was measured with four different tasks: syllable awareness, rhyme awareness, onset awareness, and tone awareness. Syllable awareness was measured with a syllable deletion task. This task was identical to the syllable deletion measure in Shu et al. [14]. There were 16 items, half of them were real words and the other half were nonsense words, all consisting of two-syllable words. Children were orally presented with each word and asked to take away the first or the second syllable. An example is: “老虎 ([lǎo hǔ], tiger), can you take away the first syllable?” The score was the number of correct answers. The Cronbach’s alpha of this task was 0.93, which is excellent.

Onset-end rhyme awareness was measured with a rhyme detection task and an onset detection task. The rhyme detection task was adapted and revised from the rhyme detection task as described in Shu et al. [14]. The task in that study had a low reliability, and hence we attempted to add two choices in each item. In our version of the task, there were two practice items and 10 test items. Children were presented with a picture of the target word first and the experimenter made sure the child gave the right word corresponding to the picture, then children were shown four choices corresponding to pictures and asked to choose the correct word that rhymed with the target word from the four choices. To make the task simple, the tone was kept constant across all the words in each item.

The onset detection task was based on the onset task used in Shu et al. [14]. The task included 2 practice items and 10 test items. The testing was similar with the rhyme detection task. Children had to choose the word that has the same onset with the target word. The total score was the number of the correct answers.

The reliability coefficients of both rhyme awareness and onset awareness tasks were low. Actually Shu et al. [14] also found a low reliability of Chinese onset and rhyme awareness tasks (0.28 and 0.49). To improve the reliability, we combined the two tasks, since they both tap into rhyme detection ability. After deleting five onset detection items and two rhyme detection items, the reliability of this task was increased to a Cronbach’s alpha of 0.60.

Tone awareness was measured with a tone discrimination task. This task was based on the tone detection task [14]. It consisted of two practice trials and 10 test trials. Each trial consisted of a pair of one-syllable words. The rhyme was the same across all the one-syllable words for this task. Children were orally presented with two words and asked if the tones of these two words were the same or different. The reliability of this task was low (0.36, which is similar to the reliability of 0.37 as reported by Shu et al.). To solve this problem, we used d-prime analysis to calculate each participant’s tone detection. In the discrimination task, % correct on the different pairs alone is not a very meaningful measure of discrimination. It becomes meaningful when interpreted in terms of the listener’s response bias, or tendency to respond “same” or “different”. Signal detection theory attributes responses to a combination of sensitivity and bias. Sensitivity is conceived as detecting a signal, which means the presence of a difference is detected and the responses to the same pairs are used as an indication of response bias. Then according to the signal detection theory, the d’ of all the participants were calculated. The d’ of 23 children were under or equal to zero and the d’ of 57 children were higher than zero. This means that 57 children were sensitive to tone discrimination. In the hierarchical regression analysis, the tone dummy variable was used to replace tone awareness. “1” was used for children who were sensitive to tone discrimination, “0” was used for children who were not sensitive to tone discrimination.

#### 2.3.5. Pinyin Letter Knowledge

Pinyin letter knowledge was measured by a Pinyin sound task. Forty-seven lower-case Pinyin letters were randomly arranged in rows on one card and children were asked if they knew their sounds. Each item was scored as correct (1) or incorrect (0). The maximum score is 47. Since all the participants were in kindergarten, none of them had received formal Pinyin education. Most of them scored at floor at this task (48 had a score of 0, 12 scored between 1 and 3). The scores of Pinyin letter knowledge which were higher than the mean score (M = 3.39, SD = 6.79) were labeled as 1 and those which were lower than the mean were labeled as 0.

#### 2.3.6. Visual Perception Skills

Visual discrimination and visual-spatial relationships subtests from Gardner’s [58] Test of Visual-Perceptual Skills Revised were administered to measure children’s visual skills. Each task consisted of one practice item and 16 test items.

The visual discrimination subtest tests the ability to match exact characteristics of a target form alongside five alternative visually similar forms. Both the target and the five geometric forms were printed in black and white. Children were asked to select the form that is identical to the target one from the five choices. The task was terminated when the child failed four out of five consecutive items. The reliability coefficient for this task was good, with a Cronbach’s alpha of 0.80.

The task of visual-spatial relationships assesses the ability to discriminate a single form or part of a single form presented in a different way from that of the other four forms of identical configuration. Each trial consisted of five black-and-white line drawings. The child had to distinguish this form from the others. The ceiling on this task was four out of five items incorrect. Cronbach’s alpha of this task was 0.90.

#### 2.3.7. Chinese Character Recognition

Chinese character recognition was identical to the character recognition task in Shu et al. [14]. It consisted of 60 single-character words which were chosen from the first grade textbooks; children were asked to read from the beginning of the test and stopped after ten consecutive mistakes. The characters in the test were listed in an increasing level of difficulty. The score was the number of correct answers.The Cronbach’s alpha of this task was 0.98.

## 3. Results

### 3.1. Descriptive Statistics

Means, standard deviations, skewness, and kurtosis for all the measures are shown in Table 1. The data of syllable awareness and visual-spatial relationships were not normally distributed. The average score of the tasks was at ceiling, and the distribution of these two tasks was negatively skewed. As recommended by Field [59], reverse score transformation was used to transform these data from negative skewed to positive skewed, then reciprocal and log transformation were used to transform them to normal distribution separately. As already mentioned in the Methods section, many children scored at floor at the Pinyin letter knowledge task. We therefore recoded this variable into a dummy score with 0 for those scoring below the mean, and 1 for those scores above the mean.

Table 2 shows the correlations among all the measures for the participants. The results showed that Chinese character recognition significantly correlated with syllable awareness and visual-spatial relationship tasks, and also correlated with vocabulary, tone awareness, and visual discrimination task. However, it did not correlate with any other measures, such as rapid naming, short term memory, and Pinyin letter knowledge. Syllable awareness significantly correlated with short term memory and the two visual perception skills, as well as with onset-end rhyme awareness and tone awareness. Onset-end rhyme awareness significantly correlated with the two visual perception skills and tone awareness. Pinyin letter knowledge correlated significantly only with tone awareness. The two visual perception skills both significantly correlated with vocabulary, and correlated with short term memory; however, they did not correlate with rapid naming and Pinyin letter knowledge.

### 3.2. Prediction of Chinese Character Recognition

In order to investigate the unique variance of Chinese character recognition accounted for by phonological awareness, Pinyin letter knowledge, and visual perception skills, we conducted hierarchical multiple regression analyses with Chinese character recognition as the dependent variable. The results are shown in Table 3.

In the first step, vocabulary, rapid naming, and short term memory were introduced as control variables. Vocabulary was entered to control for any contribution of verbal IQ. Rapid naming and short term memory were entered as phonological processing tasks. Although vocabulary, rapid naming, and short term memory accounted for 10% of the variance, they did not significantly predict Chinese character recognition.

Next, syllable awareness, tone awareness, and Pinyin letter knowledge were included in the second step. Only the two tasks of phonological awareness were entered in the regression analysis, because onset-end rhyme awareness was not significantly correlated with character recognition. Introducing the two tasks of phonological awareness and Pinyin letter knowledge explained an additional 17.1% of variance in Chinese character recognition and this change in R² was significant, *F* (6, 73) = 4.45, *p* < 0.01. Tone awareness (ß = 0.33, *t* (73) = 3.12, *p* < 0.01) and syllable awareness (ß = 0.28, *t* (73) = 2.53, *p* < 0.05) were significant predictors of Chinese character recognition.

Finally, the two visual perception skills were entered in the model. They did not improve the prediction of Chinese character recognition when entered as the third step (ΔR^2^ = 0.02). A second model with visual perception skills entered as the second step was also examined. After controlling for vocabulary, rapid naming, and short term memory, visual perception skills did not add to the prediction of Chinese character recognition, ΔR^2^ = 0.05, *F* (2, 74) = 2.20, *p* = 0.12.

### 3.3. The Role of Pinyin Letter Knowledge

Based on the results, it seems that phonological awareness had a direct effect on Chinese character reading, whereas Pinyin letter knowledge and visual perception skills did not. Since the participants were in kindergarten, most of them had a zero score at the Pinyin letter knowledge task. We further explored whether the dummy-coded Pinyin letter knowledge moderated the relationship between phonological awareness and Chinese character recognition. Separate moderation analyses were conducted with syllable awareness and tone awareness as the independent variables, Chinese character recognition as the outcome variable, and Pinyin letter knowledge as the moderator.

Figure 1 shows that the relationship between tone awareness and character recognition was moderated by Pinyin letter knowledge, *b* = 21.53, 95% CI [9.81, 33.26], *t* = 3.66, *p* < 0.001. The model was significant, *F* (3, 76) = 53.20, *p* < 0.001, R^2^ = 0.14. Tone awareness was related to character recognition, and this relation was stronger when children had a certain level of Pinyin letter knowledge. There was no moderating effect of Pinyin letter knowledge on the role of syllable awareness in predicting Chinese character recognition.

## 4. Discussion

The present study aimed to investigate the role of phonological awareness, Pinyin letter knowledge, and visual perception skills on Chinese character recognition for young children. Specifically, it examined whether visual perception skills predicted character recognition for kindergarten children over and above phonological awareness and Pinyin letter knowledge, or the other way around. The results showed that Chinese kindergarten children rely more on phonological awareness than visual perception skills and Pinyin letter knowledge to recognize Chinese characters. Moreover, it was found that Pinyin letter knowledge moderated the relationship between tone awareness and Chinese character reading.

Results first showed that syllable awareness and tone awareness predicted Chinese character recognition with vocabulary, rapid naming, and short term memory statistically being controlled for. This result corresponds with the previous findings from Shu et al. [14] and highlights the importance of both tone awareness and syllable awareness for character acquisition in a combined group of children at kindergarten level in mainland China. These results are also in line with our expectations. Syllabic units are universal in cross-language lexical processing [20,60], whereas tone can be considered integral to lexical processing in Chinese [20]. However, onset-end rhyme awareness was not significantly correlated with Chinese character recognition in the current study. This was also found in the studies of McBride-Chang et al. [20,27]. Our finding conforms with the notion that children do not make a cognitive connection between phonemes and Chinese orthography.

Second, Pinyin letter knowledge did not predict Chinese character recognition. Lin et al. [17] followed K-3 children for one year and they found that only invented Pinyin spelling emerged as a unique predictor for Chinese word reading longitudinally, whereas Pinyin letter knowledge did not. The reason may be that invented Pinyin spelling skills tap children’s phonological sensitivity, which promotes character reading [27]. One study conducted by Wang et al. [42] found that Pinyin letter knowledge was strongly related to Chinese character reading in a K-2 and K-3 combined group. However, the authors controlled only age, nonverbal IQ, and morphological awareness and not other relevant phonological skills, which we did in the current study.

With respect to the relevance of visual perception skills for learning to read in Chinese, visual discrimination and visual-spatial relationships were positively correlated with Chinese character recognition. However, they did not uniquely predict Chinese character recognition after controlling for vocabulary, RAN, and short term memory, either when phonological awareness measures were taken into account or not. The positive correlations between visual perception skills and Chinese character reading were also found in the study of Luo et al. [51]. Luo et al. [51] distinguished the respective contributions of geometric-figure processing and character-configuration processing to Chinese reading. In their study, they found that visual discrimination predicted reading in kindergarten, whereas we did not find these results in the current study. Differences in the design of the studies regarding control variables might explain the different outcomes, but more research is needed in this area.

Our results suggested that visual perception skills did not predict Chinese character reading after relevant linguistic skills were considered, at least when children had not had any formal literacy instruction. McBride-Chang et al. [50] followed Hong Kong and mainland China children for nine months in kindergarten to study the concurrent and subsequent relationship between visual-spatial relationships and Chinese character acquisition. Their results showed that the task of visual-spatial relationships uniquely predicted variance of Chinese reading only among Hong Kong children but not among mainland children at the first testing point. McBride-Chang et al. [50] confirmed that the difference between Hong Kong and mainland children may come from the fact that the formal literacy instruction begins in the first year of kindergarten in Hong Kong whereas it starts in the first year of primary school in mainland China. Visual perception skills thus seem to be relevant predictors for beginning readers, but not emergent readers, because at the beginning of learning to read, children have to attend to the salient visual features of characters [17].

Based on the previous studies and the present study, we argue that for children without reading experience, phonological awareness plays a role in Chinese character recognition. Visual perception skills may be a relevant predictor of reading skills for children who at least have some reading experience. This means that only when children acquire certain experience for character recognition, may they know how to rely on visual perception skills.

A final result that needs to be discussed is the moderating role of Pinyin letter knowledge in the association between tone awareness and Chinese character reading. Children with tone awareness who had a certain level of Pinyin letter knowledge had a better performance in character recognition task. The reason may be that although Pinyin letter knowledge is unlike advanced Pinyin skills, which usually directly capture children’s awareness of phonological sensitivity, it is still significantly correlated with advanced Pinyin skills. This was found in both the study of Lin et al. [27] and the study of Wang et al. [42]. In their results, Pinyin letter knowledge was significantly associated with invented Pinyin spelling, r = 0.60 and 0.38 respectively. Thus, the combined effect of Pinyin letter knowledge and tone awareness significantly contributed to Chinese character recognition; children with some Pinyin knowledge are better able to make the connection between speech and writing. Pinyin represents the exact sound of characters and it is reliable at the levels of the onset, rhyme, and tone of the utterance [27]. This may explain why we did not find the moderating effect of Pinyin letter knowledge on the role of syllable awareness in predicting Chinese character recognition. Syllable awareness is one kind of awareness that taps into bigger sound units when compared to tone awareness and awareness of onset and rhyme.

There are some limitations to the present study. First, the reliabilities of short-term memory and onset-end rhyme awareness were relatively low. Shu et al. [14] also found a low reliability of Chinese onset and rhyme awareness tasks. They explained the low reliability of Chinese phonological awareness tasks from the fact that children were asked to choose the answers from limited numbers of response alternatives. Although in the present study the number of alternatives in these tasks were increased, the reliabilities were still relatively low. In future studies, opened-ended methods may be used to increase the reliability of Chinese phonological awareness measures. A second limitation is that the study did not follow a longitudinal design. To arrive at more comprehensive answers to the question of how Chinese character recognition can be predicted from phonological awareness, Pinyin letter knowledge, and visual perception skills, there is an urgent need for longitudinal studies.

To conclude, the present study showed that syllable awareness and tone awareness can be seen to be highly relevant to word reading in kindergarteners even after controlling for cognitive skills in mainland China. Pinyin letter knowledge is also relevant to word reading as it can help children to make the connection between Chinese speech and writing. With an eye on classroom practice, for preschool and kindergarten caretakers, a better focus on children’s syllable awareness and tone distinction in music and language games, and attention for Pinyin letter knowledge in a literate classroom environment can be seen as an important step in helping Chinese children to accomplish the transition from phonological awareness to early literacy.

## Figures and Tables

**Figure 1 behavsci-12-00254-f001:**
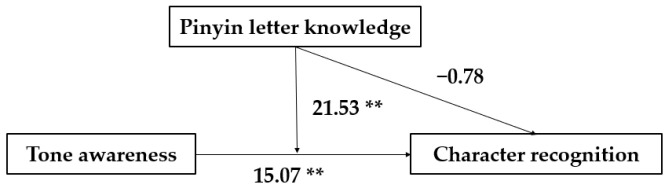
Moderation model of Chinese tone awareness, Pinyin letter knowledge, and character recognition. Note, ** *p* < 0.01.

**Table 1 behavsci-12-00254-t001:** Means, standard deviation, skewness and kurtosis of all the measures for the participants (N = 80).

Variables	M	SD	Skewness	Kurtosis
Vocabulary	83.25	21.93	−0.65	−0.43
Rapid naming	41.50	8.20	−0.83	−0.18
Short term memory	10.70	1.26	−0.70	−0.39
Syllable awareness	14.53	3.19	−0.75	−1.19
Onest-end rhyme awareness	5.08	3.06	−0.05	−0.51
Tone awareness	6.33	1.83	−0.43	0.97
Pinyin letter knowledge	3.39	6.79	2.45	5.61
Visual discrimination	9.35	3.66	−0.70	0.32
Visual-spatial relationships	10.74	4.63	−0.11	−0.49
Character recognition	31.34	17.71	−0.09	−1.32

Note: The original data of tone awareness and Pinyin letter knowledge was included in Table 1.

**Table 2 behavsci-12-00254-t002:** Correlations between vocabulary, two phonological processing skills, phonological awareness, Pinyin letter knowledge and visual perception skills and character recognition (N = 80).

	1	2	3	4	5	6	7	8	9	10
1. Vocabulary	-									
2. Rapid naming	0.39 **	-								
3. Short term memory	0.18	0.19	-							
4. Syllable awareness	0.15	0.07	0.42 **	-						
5. Onset-end rhyme awareness	0.13	−0.07	0.17	0.28 *	-					
6. Tone awareness	0.05	−0.11	0.07	−0.001	0.42 **	-				
7. Pinyin letter knowledge	−0.01	0.08	0.16	0.03	0.07	0.24 *	-			
8. Visual discrimination	0.26 *	0.05	0.28 *	0.30 **	0.30 **	0.08	0.06	-		
9. Visual-spatial relationships	0.40 **	0.20	0.27 *	0.37 **	0.42 **	0.15	−0.04	0.51 **	-	
10. Character recognition	0.28 *	0.19	0.17	0.30 **	0.19	0.33 **	0.14	0.25 *	0.33 **	-

Note. ** *p* < 0.01, * *p* < 0.05. Tone dummy variable and Pinyin dummy variable are included in Table 2.

**Table 3 behavsci-12-00254-t003:** Results of the hierarchical regression analysis in predicting Chinese character recognition after controlling for age, vocabulary, rapid naming, and short term memory (N = 80).

Step	Predictor	Δ R²	*B*	*SE*	ß
1	Vocabulary	0.10	0.18	0.10	0.23
	Rapid naming		0.17	0.26	0.08
	Short term memory		1.61	1.58	0.11
2	Vocabulary	0.17 **	0.14	0.09	0.17
	Rapid naming		0.30	0.24	0.14
	Short term memory		−0.50	1.60	−0.04
	Syllable awareness		14.23	5.62	0.28 *
	Tone awareness		12.71	4.08	0.33 **
	Pinyin letter knowledge		2.02	4.27	0.05
Total	R^2^adj	0.21 **			

Note. * *p* < 0.05; ** *p* < 0.01.

## Data Availability

The data presented in this study are available on request from the corresponding author.

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
