# Peer review of "The Role of Phonological Awareness, Pinyin Letter Knowledge, and Visual Perception Skills in Kindergarteners’ Chinese Character Reading"

_behavsci, 2022, doi:10.3390/bs12080254_

Round 1
Reviewer 1 Report
Please see the attached file.

Author Response
Thank you very much for the positive review.
In line 135, what do the authors mean by “Time 1 syllable deletion?”
Response: We mean the syllable deletion that was tested at Time 1. We have made it clear in the revised version.
- 2.3 (lines 351-58) on the procedure should be moved so that it comes between what are now §2.1 and §2.2. That is, §2.3 should become §2.2 and the former §2.2 should become the new §2.3
Response: We have followed your comments and changed the orders of the three parts.
Idiomaticness of the English
Response: We have followed your comments and revised them. Please see the revised version.
Reviewer 2 Report
This is an interesting paper that is adding crucial information needed in identifying important skills for developing reading skills. Some minor edits are required.
Wording of the sentences on the following lines should be reviewed for grammar/punctuation, word choice, completeness (missing word), or coherence among the different phrases:
- 100 – 102 (missing word?)
- 130 – 133 (missing word?)
- 144 – 148 (appropriateness of “tap” on line 147)
- 162 – 163 (grammar)
- 207 – 209 (punctuation)
- 217 – 218 (wording of “A number of 80”)
- 286 – 289 (grammar on “present with” and “of target”)
- 301 – 302 (wording of “task of tone detection task”)
- 311 – 313 (grammar – maybe divide into two sentences after “detected”)
- 316 – 319 (grammar/punctuation – maybe into two sentences after “awareness”)
- 333 – 334 (wording - “alternative five”)
- 346 – 348 (use of comma between “textbooks” and “children”)
- 353 – 356 (wording - “accord”)
- 356 – 358 (appropriateness of the word “measured”)
- 514 – 516 (grammar of “tap” and “onset, rhyme” punctuation)
The authors switch between using “rhyme” and “rime” throughout the paper. Is there a difference between the two terms? If so, the difference should be noted and described. If not, one term should be used consistently instead of switching between the two.
An extra reference (Peng et al., 2017) is included in the references section (lines 625 – 626) but not referenced in the body of the paper. The reference should be added to the body or removed from the references section.
Author Response
Comments and Suggestions for Authors
This is an interesting paper that is adding crucial information needed in identifying important skills for developing reading skills. Some minor edits are required.
Thanks for the positive review and comments.
Wording of the sentences on the following lines should be reviewed for grammar/punctuation, word choice, completeness (missing word), or coherence among the different phrases:
- 100 – 102 (missing word?)
Response: We have changed the sentence into “Moreover, some homophonic written
characters are included in a phonetic radical neighborhood and are visually similar, such
as the following three characters “妈”, “马”, and “骂”.
- 130 – 133 (missing word?)
Response: We have changed it into “For example, Wang et al. found that after age,
kindergarten level, and nonverbal reasoning statistically controlled, Pinyin letter name
knowledge, morphological awareness, and rapid naming uniquely predicted Chinese word
reading in a combined K-2 and K-3 group”.
- 144 – 148 (appropriateness of “tap” on line 147)
Response: We have changed “tap” to “involve”.
- 162 – 163 (grammar)
Response: We have revised it into “For example, in the Chinese character 妈 ‘mother,’
the components are arranged side by side. This character 妈 is composed of the radical
女 ‘female’ and another symbol 马 that represents the segmental string or syllable /ma/.
- 217 – 218 (wording of “A number of 80”)
Response: We have changed it to “A total of 80”.
- 286 – 289 (grammar on “present with” and “of target”)
Response: We have revised to “of the target”.
- 301 – 302 (wording of “task of tone detection task”)
Response: We have deleted “task of”.
- 311 – 313 (grammar – maybe divide into two sentences after “detected”)
Response: We have followed your comment and divided it into two sentences.
- 316 – 319 (grammar/punctuation – maybe into two sentences after “awareness”)
Response: We have followed your comment and divided it into two sentences.
- 333 – 334 (wording - “alternative five”)
Response: We have revised to “five alternative”.
- 346 – 348 (use of comma between “textbooks” and “children”)
Response: We have followed your comment.
- 353 – 356 (wording - “accord”)
Response: We have revised to “according”.
- 356 – 358 (appropriateness of the word “measured”)
Response: We have changed it to “conducted”.
- 514 – 516 (grammar of “tap” and “onset, rhyme” punctuation)
Response: We have changed it to “Syllable awareness is one kind of awareness that taps
into bigger sound units when compared to tone awareness and awareness of onset and
rhyme”.